# Nutritional Solution for the Italian Heavy Pig Production to Improve Nitrogen Efficiency While Maintaining Productive Performance and Meat Quality

**DOI:** 10.3390/ani15091309

**Published:** 2025-04-30

**Authors:** Sujen Eleonora Santini, Elena Zanelli, Valerio Faeti, Gianni Marchetto, Maria Teresa Pacchioli, Sara Carè, Davide Bochicchio

**Affiliations:** 1Council for Agricultural Research and Economics, Research Centre for Animal Production and Aquaculture, Via Beccastecca 345, 41018 Modena, Italy; elena.zanelli@crea.gov.it (E.Z.); valerio.faeti@crea.gov.it (V.F.); gianni.marchetto@crea.gov.it (G.M.); 2Centro Ricerca Produzioni Animali, Via Timavo 43/2, 42121 Reggio Emilia, Italy; m.t.pacchioli@crpa.it; 3Council for Agricultural Research and Economics, Research Centre for Agricultural Policies and Bioeconomy, Borgo XX Giugno 74, 06121 Perugia, Italy; sara.care@crea.gov.it

**Keywords:** Italian heavy pig, low-crude-protein diet, protein, nitrogen efficiency

## Abstract

In Italy, pig farming is mainly based on the production of Protected Designation of Origin (PDO) heavy pigs for cured ham. The aim of our study was to demonstrate new rationing models, in accordance with stringent PDO feed regulations, that achieve an equal or better performance in regard to reducing the environmental impact linked to nitrogen excretion. In particular, from 50 kg live weight (LW) to slaughter (about 170 kg LW), we compared three theses: (1) (C) traditional diet based on soy and maize; (2) (T1) iso-protein diet protein, pea-based; and (3) (T2) low-crude-protein diet, cereal-based. The results demonstrate that it is possible to obtain the same productive performance and quality characteristics of the meat while significantly (*p* < 0.01) reducing the nitrogen excretion by almost 30%.

## 1. Introduction

Italian pig farming essentially has the goal of breeding heavy pigs for production of Protected Designation of Origin (PDO) dry-cured products, such as Parma hams [1]. As a matter of fact, Italian pig breeding differs from that of other European countries because pigs intended for slaughter have a BW of 150 to 170 kg, and their production is strictly regulated (Consorzio del Prosciutto di Parma 2023 [2]). For pig production, agricultural and breeding activities take place intensively in areas that are already heavily anthropized, where the management of environmental compatibility is mandatory for the increase in public attention, as well as the development of environmental legislation. One of the major criticisms in modern pig production is nitrogen excretion, which is because of the negative impact on the environment, such as manure ammonia (NH_3_) contribution to acidification and eutrophication of sensitive ecosystems and odor emissions [3,4]. Globally, pig production represents 14% of livestock NH3 emissions and 8% of total anthropogenic NH_3_ emissions [5]; on average, feed contributes to 60% of the climate change impact of pig production, and manure emissions contribute 30% [6]. For this reason, the European Directive 91/676/EC, aiming to prevent or reduce the nitrate pollution of surface and underground water, stated that each member state must identify the vulnerable areas where the load of N of livestock origin cannot exceed 170 kg/ha/year. Consequently, to avoid the reduction of livestock unit/ha, several strategies have been proposed in which nutritional solutions are a major lever, as they can target the two main impact sources: feed and manure.

Lowering dietary crude protein (CP) content reduces the deamination of excess AA, as well as N excretion [7], which would reduce the environmental pollution of N and be beneficial for the climate. According to ERM/AB-DLO methodology [8], N excretion is quantified as the difference between N consumption and N retention in animal products. However, as ERM reference values have been defined mainly on the basis of North Europe, the Italian government supported the “Interregional project on N excretion from livestock”, involving the Northern Italy regions, where PDO pig production take place, aimed to quantify N excretion and identify the source of variation for the main national husbandry systems [8]. The results are used as reference values to set up regional or national regulations. Nitrogen excretion decreases linearly with decreasing CP, quantified at approximately 8% per point of CP reduction, while NH_3_ emissions are reduced by 10% per point of CP reduction [9,10,11,12].

Consequently, dietary CP reduction with free amino acid (AA) supplementation has been suggested by several authors in the last few years as a means to still achieve acceptable performance, carcass characteristic, and meat quality [13,14,15,16,17,18], and it has been promoted for several decades in Europe to reduce N emissions and the import of soybean meal associated with deforestation [19,20]. In practice, the reduction of the dietary protein concentration by 3% units of the NRC (1998) [21] decreased total N excretion by an average of 24.5% without adversely affecting the growth performance and carcass quality of heavy-growing finishing pigs [22]. However, conflicting results are reported on the influence of a further reduction in dietary CP level on growth performance [23,24,25], and, in general, few experiments have tested the effects of very low CP levels on performance, N balance, and emissions, above all, in heavy pigs.

In this context, last September 2024, the new production regulations for Protected Designation of Origin “Parma Hams” came into force, which revised various aspects of the production system, including greater flexibility in the formulation of the diet with the aim of allowing ration models, which, with the same production efficiency and quality, allow the environmental impact to be reduced. For example, the minimum level of the percentage of proteins in finished feed has been removed (which was set at 12%), and it has been established that at least 50%* of the feeding raw materials must come from the geographical area of farming. Furthermore, the new production regulation promotes the use of other protein crops, such as protein pea, whose maximum limit has been increased from 5% to 25%* (* values refer to the dry matter of the finished feed).

Regarding feeding, in Italy, in 2020, about 9 million tons of maize were used for animal feed, compared to 1.2 million tons of soft wheat, 1.0 million tons of barley, and 0.6 million tons of other cereals (National Association of Livestock Food Producers (ASSALZOO). According to the same reference, 3.9 million tons of soy oilcake and soybean oil meal were used out of a total of 4.8 million tons derived from oilseeds; of the 3.9 million tons, only 0.6 million was derived from the Italian production of soybeans. Consequently, maize and imported soybean oil meal are currently the main raw materials used in heavy-pig feed and, in general, in livestock feed. About maize, it is necessary to consider two potential critical points. The first point we must consider is its concentration of linoleic acid. To avoid soft and oily ham subcutaneous fats, which may turn rancid during the 12–16-month curing process, the regulations severely restrict the use of added fats and high-fat feedstuffs in the diets of finishing heavy pigs, pointing out that the linoleic acid should not exceed 2% of the diet (on a dry-matter basis). Hams whose subcutaneous fat shows a high unsaturation level (more than 15% linoleic acid on total fatty acids) are not admitted to the PDO raw ham production. The second point to consider is the large amounts of irrigation water that are provided to maximize yields [26]; particularly because of global climate change [27], that might lead to a decrease of water availability for irrigation [28]. For this reason, it will be increasingly important to take into consideration less water-demanding crops, such as barley and sorghum [29]. Previous studies [30,31] showed that diets based on barley might be suitable for the heavy-pig breeding intended to produce Italian PDO products; also, sorghum is suitable mainly for their resistance to water stress and risk of mycotoxins, thus making sorghum generally safer than maize in this respect. It is also interesting to note that cereals with a particularly high protein content could replace the soybean oil meal in the feed of heavy pigs, allowing Italy to reduce its dependence on foreign countries for livestock feed and, thus, leading to a decrease in greenhouse gas emissions and allowing Italy to meet the PDO request for feed raw materials to have a geographical origin. Indeed, in the diet of heavy pigs for PDO, both barley and sorghum can individually be included as up to 55% of the dry matter of the ration.

Taking all the premises into account, the aim of this study was to evaluate new rationing models, according to the PDO request, with alternative raw materials to maize and soybean oil meal, and with lower protein content than traditional materials, to test and demonstrate their technical efficiency and environmental sustainability in terms of nitrogen efficiency. The results show that it is possible, by improving the efficient use of proteins and using alternative sources of protein, to obtain the same performance (as average daily growth and feed conversion rate) and quality characteristics of heavy-pig meat, compared to a traditional ration, significantly reducing (*p* < 0.01) nitrogen excretion by almost 30%.

## 2. Materials and Methods

### 2.1. Animals and Feeding Thesis

The trial involved 72 pigs (Italian Duroc × Italian Large White) from 55.41 ± 1.62 kg (mean ± SD) live weight, divided homogeneously in 18 boxes of 4 animals according to the following feeding theses: 6 boxes were fed with traditional diet as control (C) rations, 6 boxes were fed with protein pea-based diet (T1), and 6 boxes were fed with cereal-based diet (T2), as shown in the tables below (Table 1a,b).

Three different diets were formulated for each thesis according to the animals’ live weight, balanced for the ratio of lysine D/digestible energy, and the rations were calculated to provide the animals with the same energy and amino acid intakes. Aminogram was formulated according to J. van Milgen’s ideal protein concept [32].

### 2.2. Data Collection and Calculation

All pigs were weighed every 28 days until the slaughter weight was reached after 139 days of trial, at an average live weight of 173.9 ± 2.74 (mean ± SD) kg. The animals were fed with rationed feeding according to specific needs. Based on feed ingestion and live body weight, average daily growth (ADG) and feed conversion rate (FCR), as productive parameters, were calculated. Additionally, according to G. Xiccato’s method [9], we evaluated nitrogen (N) excretion and efficiency during the production cycle considered, represented as the following equations:N excretion (g) = [N intake (g) + N animal IN (*)] − N animal OUT (*)N efficiency (%) = {[N animal OUT (*) − animal IN (*)]/N intake (g)} × 100
where (*) N animal IN and OUT represent the quantity of nitrogen present in the animals according to a fixed coefficient per weight category [8], as shown below (Table 2).

After slaughter, a series of data were collected on the carcasses following the European directive (2014/38/EU) relating to the authorization of pig carcass classification methods in Italy. The cold weight is obtained by deducting 2% of the hot weight of the carcass measured within the first 45 min of the animal’s death and refers to the bled and emptied carcass (EU Regulation n. 1308/2013). The slaughterhouse carried out the evaluation with the Fat-O-Meat’er 2 instrumentation (Frontmatec, Kolding, Denmark) from which the fat value corresponds to the thickness of the fat in mm and the meat value to the thickness of the loin. The percentage of lean meat is evaluated according to the FOM2 equation; pH of the semimembranosus and biceps femoris muscles at 45′ after slaughter was measured in the thigh. Twenty-four hours after slaughter, the thighs were weighed to calculate the cooling drop, the pH of the semimembranosus and biceps femoris were measured, and the color of the same muscles and cover fat were measured with a Minolta colorimeter according to the CIE L*a*b* system [33]. A full-thickness fat cover sample was taken from one thigh per pig for fatty acid composition analysis.

### 2.3. Fatty Acids Analysis

To verify the suitability of backfat for seasoning, the fatty acid composition of subcutaneous fat was analyzed. Following De Pedro et al. [34], the subcutaneous tissue was chopped into small pieces (approximately 0.5 cm^3^), placed in individual microwave-resistant glass containers, and heated in a laboratory microwave oven of 650 W power and 2450 MHz microwave frequency for 10 min at 33% power. The liquid fat obtained was filtered through anhydrous sodium sulphate. These samples were used for fatty acid composition analysis. The preparation of methyl esters of subcutaneous fat was performed following the method proposed by Stoffel et al. [35]. About 100 mg of fat was dissolved in methanolic HCl (5 mL) and placed at 100 °C for 40 min. After cooling, 2 mL of water and 2 mL of hexane were added. The organic phase was filtered through anhydrous sodium sulphate and injected into a GC 2010 pro gas chromatograph (Shimadzu, Kyoto, Japan) equipped with a flame ionization detector and an automatic injection system (AOC-20i). The column was a SP 2380 fused-silica capillary column (30 m length, 0.25 mm i.d. 0.20 mL film thickness). The initial column temperature was 140 °C, and it was raised to 250 °C in 27 min. The injector and detector temperatures were at 250 °C. The helium carrier gas pressure was 143 kPa, and the split ratio was 1:100. LabSolutions software for Windows was used for data analysis. Identification was accomplished by comparing the retention time of the unknown fatty acid methyl esters (FAMEs) with those of known FAME standard mixtures.

### 2.4. Statistical Analysis

The statistical analysis for the feed trial data was performed by box, with 6 repetitions per thesis (3 boxes of 4 animals for each sex), while the postmortem data were analyzed for each individual animal (72 animals). All data were analyzed through the analysis of variance with the GLM procedure of SAS version 9.4 for Windows (SAS Institute Inc., Cary, NC, USA). The means value was estimated with the LSMEANS procedure of the SAS GLM. Variables that differed by *p* ≤ 0.05 were tested with the Bonferroni post hoc test, with alpha = 0.05 and alpha = 0.01. The live results were processed, taking repetition as the minimum cell (4 repetitions per thesis), with analysis of variance according to the following model:Yijk = M + Ai +Bj + (AB)ij Eijk
where
Yijk = dependent variable observed on the kma replication of the ijmo subgroup;M = overall mean;Ai= diet (i = 1,3);Bj = sex (j = 1.2);ABij = interaction thesis x sex;Eijk = residual error.

Where the mean initial sex factor was considered to be the blocking factor.

Slaughter results were processed, taking the individual animal as the minimum cell with analysis of variance according to the following model:Yijk = M + Ai +Bj + (AB)ij + Eijk
where
Yijk = dependent variable observed on the kmo individual of the ijmo subgroup;Ai = diet (i = 1,3);Bj = sex (j = 1,2);ABij = interaction thesis x sex;Eijk = residual error.

Averages were calculated through LSMEANS of SAS GLM procedures.

Variables that differed by *p* ≤ 0.05 were tested by Bonferroni’s test with alpha = 0.05 and alpha = 0.01.

## 3. Results

### 3.1. Productive Performances

The feed ingestion (kg/animal) measured during the experimental period is reported in Table 3.

No significant difference was found in the final weight at slaughter and in the productive parameters of average daily growth and feed conversion rate (Table 4).

### 3.2. Nitrogen Excretion and Efficiency

The nitrogen balance indicates a greater efficiency of nitrogen use in the cereal-based (T2) diet thesis (Table 5). In particular, the N excretion is significantly (*p* < 0.01) lower (−28%) for the cereal-based diet (T2) compared to both the traditional (C) and protein pea-based diets (T1). Coherently, N efficiency is also significantly (*p* < 0.01) higher (+21%) for the T2 diet compared to both the C and T1 diet theses.

### 3.3. Postmortem Data

The statistical analysis of all data collected after slaughtering revealed no significant difference between the theses (Table 6 and Table 7).

The linoleic acid of the backfat of pigs fed the C diet is significantly higher than that present in pigs in the T1 diet, but not than that in the T2 diet (C 9.928 a vs. T1 9.113 b vs. T2 9.404 ab, *p* < 0.05).

Table 7 shows the statistical analysis of the averages of the fatty acid content of the backfat of pigs fed with different diets.

## 4. Discussion

The pig sector in Italy is peculiar in respect to other European countries, since it is driven by the production of certified (PDO) cured ham. The heavy live weight (160 kg or more) and the minimum age (9 months) at slaughter imposed by PDO regulation strongly affect feeding practices and diet characteristics, which could also have negative impacts on the environmental sustainability of production. The adoption of reference coefficients (which is based on 25% of N retention efficiency) for computing farm N excretion based exclusively on the number of heads would not promote the on-farm application of strategies aimed at reducing the excretion per head. This kind of approach simply acts to reduce the number of livestock unit/ha, with serious risks of a strong decrease in the number of livestock farms, particularly in the areas vulnerable to nitrate pollution. In this contest, although the benefits of reduction of dietary CP for the environment due to lower nitrogen excretion are well established [36], the economic efficiency of low-CP diets may vary with the availability and price of ingredients and pig performance [11,37,38,39]. Therefore, the influence of dietary CP level on growth performance (such as ADG and FCR), carcass characteristic, and meat quality is included as an important criterion to justify low-CP diets. As for pigs, the need for dietary protein is essentially a need for amino acids (AAs) [40], and the nitrogen utilization efficiency of swine varies based on protein source, which may be related to AA release kinetics [41,42,43]. Recent findings showed that the kinetics of AA released from dietary protein have substantial effects on protein synthesis in muscle, suggesting that AA release kinetics are closely related to nitrogen deposition in animals [44]; this aspect could be especially relevant in low-nitrogen diets due to the excessive use of individual AAs to optimize protein content. The protein pea-based (T1) ration demonstrates that it is possible to change the protein source and maintain the same CP values without affecting production and quality parameters. Based on a recent review [45], there is a minimum crude protein level after which the growth performance of pigs can be compromised, even though diets are balanced for all essential AAs. The standardized ileal digestible (SID) lysine (Lys)/CP ratio has been proposed as a way to evaluate the intensity of CP reduction [20]. In particular, Millet et al. reported that only a few experiments have explored CP reduction in the range of 6.2 to 7.0% SID Lys:CP, which may result in impaired performance that would compromise the environmental benefits expected from the reduction of N excretion and subsequent emissions. In addition, Henry and Dourmad [46] reported that a limit value of 6.5% for the total Lys/CP ratio can be used for fattening pigs: above this value, which is close to the “ideal” protein ratio, the total protein intake may be insufficient in itself.

Our experimental cereal-based diet (T2) explored the reduction of CP in the range of 6.3 to 5.0% of SID Lys:CP corresponding to the range of 7.4 to 6% of total Lys:CP ratios. Consequently, the low CP T2 ration in the finishing phase (120–170 kg body weight), which is a typical phase of heavy pigs, achieves a reduction in CP greater than the limit values reported in the literature. In particular, according to our results, a 24.5% reduction (as an average on the considered period from 50 kg BW to slaughter) of CP of the cereal-based diet (T2) compared to the traditional diet (C) did not influence the parameters of production efficiency and quality considered. Instead, from an environmental point of view, this reduction in CP intake and, consequently, in N ingestion resulted in a significant reduction in excreted N, equal to −28% and −27.6% compared to diets C and T1, respectively; moreover, the efficiency of nitrogen was significantly greater than 21% compared to both diets C and T1. As previously mentioned in Table 6, the result of the % of lean meat of the carcass was consistent with the calculations of the nitrogen efficiency, thus confirming that the T2 diet had a significantly better nitrogen efficiency than the C and T1 rations.

Protein is a relatively expensive nutrient, and due to the increase in feed intake during the finishing phases, the efficiency of nutrient use greatly impacts the costs incurred by the production system [32]. Thus, the application of low-CP diets can also be considered a cost-effective alternative feeding strategy due to the higher price of protein ingredients versus energy sources. In addition, low-protein nutritional strategies appear to have a direct impact on the immune status of pigs [47] and have potential use to prevent or treat stress-related health problems, including heat stress [48].

Regarding the use of maize in heavy-pig feeding, it must be considered that PDO cured-ham disciplinary codes fix a maximum level of linoleic acid in the pig diet at 2% on dry-matter intake. As Table 1b shows, the amount of linoleic acid in the ration is mainly related to the amount of maize. In fact, PDO cured-ham disciplinary codes established a maximum limit of 65% on a DM basis for the use of maize in pig diets. The limit is related to the high content on linoleic acid (C18:2) in maize fat, which can negatively affect fat quality in the finished product by increasing its unsaturation level. Fat with a high unsaturation level is more exposed to lipid oxidation, is less compact and white, and therefore has undesirable fat characteristics according to customers [49,50]. Therefore, the percentage of linoleic acid in the pigs’ subcutaneous fat is related to the amount of linoleic acid contained in the diet. In this trial, according to Table 6, the percentage of linoleic acid in subcutaneous fat is well below the limit of 15% imposed by the PDO cured-ham disciplinary codes. According to Della Casa [51], a small difference in linoleic acid content in diet can lead to significant differences in the fatty acid composition of depot fats.

Taken together, our results show that it is possible, even for heavy pigs, to adopt low-CP nutritional strategies whose application is potentially decisive for the main aspects that will influence the sustainability of future farming: production efficiency, environmental impact, and animal welfare.

The methodological approach followed in this study to calculate N excretion and efficiency can be used at the farm level, where the adoption of protocols for recording feed consumption, feed composition, and production performance would promote a better identification of the critical points of husbandry and a more accurate quantification of excretions, as well as the need of agricultural land.

## 5. Conclusions

Taken together, these data indicate that the use of different raw materials (such as barley, sorghum, and protein pea) and the reduction of the total protein content did not influence the technical efficiency and quality parameters even when the formulations go over the minimum level indicated as ideal protein. In addition, using AA supplementation makes it possible to use a broader range of protein sources in the diet and lower the dependency on soybean meal, of which the production can have negative effects on the environment and climate [52]. Otherwise, a highly significant difference (*p* < 0.01) was recorded in the nitrogen efficiency of the cereal-based diet both in terms of N excretion and N efficiency, highlighting the importance to promote feeding strategies capable of increasing N retention per animal product unit, as well as the methodology to measure that at farm level. These results open up very interesting scenarios on the future feeding of heavy pigs and the cropping systems on which this is based.

## Figures and Tables

**Table 1 animals-15-01309-t001:** (**a**) Feeding thesis as composition. (**b**) Feeding thesis as nutrients.

(a)									
	**C**			**T1**			**T2**		
**LW (kg)**	**P1 ^1^**	**P2 ^2^**	**P3 ^3^**	**P1**	**P2**	**P3**	**P1**	**P2**	**P3**
Composition									
Maize (%)	45	47.5	53	-	-	-	18	33.7	28
Barley (%)	30	31	30	32	24.7	20.6	17	15	10
Sorghum (%)	-	-	-	28	37	50	50	37.6	50
Soybean oil meal (%)	10.5	7.9	5.3	-	-	-	-	-	-
Protein pea (%)	-	-	-	25	25	17.2	-	-	-
Bran (%)	10	9	8	10	9	8	10	9	8
Lard (%)	1.6	1.5	1	1.5	1.4	1.3	1.4	1	0.7
Lysine (%)	0.27	0.23	0.20	0.21	0.10	0.12	0.66	0.54	0.40
Methionine (%)	0.03			0.10	0.04		0.14	0.08	
Tryptophan (%)				0.01			0.05	0.03	
Threonine (%)	0.05	0.03		0.05			0.21	0.15	0.08
**(b)**									
	**C**			**T1**			**T2**		
**LW (kg)**	**P1 ^1^**	**P2 ^2^**	**P3 ^3^**	**P1**	**P2**	**P3**	**P1**	**P2**	**P3**
Digestible energy (Kcal)	3250	3250	3240	3230	3250	3240	3230	3230	3230
Protein (%)	14	13	12	14	13	12	10.8	10.5	10
Lipids (%)	4	4	3.7	3.5	3.5	3.5	4	3.7	3.5
Fiber (%)	3.6	3.4	3.1	4.1	3.9	3.5	3	2.7	2.7
Lysine (%)	0.8	0.7	0.6	0.8	0.7	0.6	0.8	0.7	0.6
SID lysine (%)	0.68	0.59	0.5	0.67	0.58	0.5	0.68	0.59	0.5
Starch (%)	45	47	49.6	47	48.7	51.3	53.5	54.3	56
Linoleic acid (%)	1.65	1.69	1.73	1.11	1.14	1.20	1.54	1.67	1.61

^1^ P1 = starter from about 50 to 90 kg BW, ^2^ P2 = grower from about 90 to 120 kg BW; ^3^ P3 = finisher from about 120 to 170 kg LW.

**Table 2 animals-15-01309-t002:** Fixed coefficient per live weight category representing the quantity of N present in the animals [9].

Live Weight (kg)	g N/kg Live Weight
40–80	26
80–120	25
120–170	24

**Table 3 animals-15-01309-t003:** Feed ingestion (kg/animal) measured during experimental period.

Feeding Phase	Day	Feed Ingestion (kg/Animal)
P1 ^1^	42	88.1
P2 ^2^	44	133
P3 ^3^	53	185.5
total	139	406.6

^1^ P1 = starter from about 50 to 90 kg BW, ^2^ P2 = grower from about 90 to 120 kg LW; ^3^ P3 = finisher from about 120 to 170 kg LW.

**Table 4 animals-15-01309-t004:** Productive performance for diet theses.

Variable	C	T1	T2	*p*
Initial live weight (kg) ^1^	56.06 ± 1.24	55.13 ± 2.12	55.05 ± 1.47	0.41
Final live weight (kg) ^1^	174.30 ± 1.71	174.61 ± 2.55	172.8 ± 3.72	0.37
ADG ^2^ (kg)	0.851 ± 0.01	0.860 ± 0.02	0.847 ± 0.02	0.32
FCR ^3^	3.47 ± 0.06	3.42 ± 0.06	3.49 ± 0.08	0.37

^1^ The live weight is expressed as mean ± SD; ^2^ average daily growth; ^3^ feed conversion rate.

**Table 5 animals-15-01309-t005:** Nitrogen (N) excretion and efficiency during the production cycle considered.

Variable	C	T1	T2	*p*
N input per animal (g)	1457.52	1433.47	1431.19	0.49
N intake (g)	8301.44	8301.44	6724.76	-
N output per animal (g)	4183.10	4190.70	4147.10	0.36
N excretion (g)	5575.86 ^A^	5544.21 ^A^	4008.86 ^B^	<0.0001
N efficiency (%)	0.33 ^B^	0.33 ^B^	0.40 ^A^	<0.0001

^AB^ Different letters within a row indicate significant differences (*p* < 0.01).

**Table 6 animals-15-01309-t006:** Postmortem data collecting.

Variable	C	T1	T2	Female	Male	*p*
Cold weight (kg)	141.4	141.9	140.9	140.8	142.1	0.99
Fat (mm)	27.8	26.0	29.1	27.6	27.7	0.43
Meat (mm)	63.4	68.3	64.9	65.5	65.5	0.54
% lean meat	53.0	54.0	52.6	53.2	53.2	0.41
pH45 Semimembranosus	6.35	6.28	6.26	6.28	6.32	0.69
pH45 biceps femoris	6.34	6.26	6.33	6.28	6.34	0.57
pH24 semimembranosus	5.77	5.67	5.68	5.70	5.71	0.11
pH24 biceps femoris	5.82	5.75	5.76	5.77	5.78	0.20
L semimembranosus	44.85	45.81	45.57	46.07	44.75	0.54
A semimembranosus	1.93	2.02	2.71	2.05	2.39	0.15
B semimembranosus	8.44	8.44	8.62	8.41	8.59	0.78
L biceps femoris	46.47	44.70	48.19	46.94	45.98	0.18
A biceps femoris	3.37	4.05	3.87	3.83	3.70	0.43
B biceps femoris	8.98	9.06	9.51	9.33	9.04	0.42

**Table 7 animals-15-01309-t007:** Ham’s subcutaneous-fat fatty acid composition.

Fatty Acids ^1^	C	T1	T2	F	M	*p* General Model	*p* Thesis	*p* Gender	*p* THESIS X Gender
C14:0	1.183	1.170	1.170	1.184	1.164	0.0292	NS	NS	0.0041
C16:0	22.011	21.648	22.068	22.216 ^A^	21.602 ^B^	0.0054	NS	0.0027	NS
C16:1	2.277	2.290	2.318	2.266	2.323	NS	NS	NS	NS
C18:0	14.092	16.275	13.947	15.596	13.947	NS	NS	NS	NS
C18:1	48.497	47.566	49.054	47.300	49.445	NS	NS	NS	NS
C18:2	9.928 ^a^	9.113 ^b^	9.404 ^ab^	9.528	9.435	0.0289	0.0029	NS	NS
C18:3	1.398	1.420	1.416	1.338 ^b^	1.484 ^a^	0.017	NS	0.0017	NS
C20:0	0.486 ^a^	0.435 ^b^	0.451 ^ab^	0.439	0.476	0.0146	0.0441	0.0284	NS

^AB^ Different letters within a row indicate significant differences (*p* < 0.01). ^ab^ Different letters within a row indicate significant differences (*p* < 0.05). ^1^ Expressed as total fatty acid percentage.

## Data Availability

The original contributions presented in this study are included in the article. Further inquiries can be directed to the corresponding authors.

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
