# Peer review of "Nutritional Solution for the Italian Heavy Pig Production to Improve Nitrogen Efficiency While Maintaining Productive Performance and Meat Quality"

_animals, 2025, doi:10.3390/ani15091309_

Round 1
Reviewer 1 Report
Comments and Suggestions for Authors
I would like to thank the researchers for a very important study in the field of animal nutrition.
I think it would be better if the suggestions tagged as comments in the manuscript file were taken into account.

Author Response
thanks for the valuable and useful suggestions that have been applied to the new version of the text attached

Reviewer 2 Report
Comments and Suggestions for Authors
After careful evaluation of the manuscript (animals-3606317), titled “Nutritional solution in very low crude protein diets for the Italian heavy pig PDO production to improve nitrogen efficiency while maintaining productive and quality performance.” I believe there are some serious requirements for editing and rewriting this manuscript by an English-language native expert as the text contains many grammatical and technical term errors. There are also some deficiencies and therefore, the authors will require to carefully revise them for the manuscript before being considered for publication in Animals. My recommendations are provided below:
Introduction:
- The “introduction” section is too long and scattered. Please rewrite and focus on the problem definition, short explaining of the previously available literature, defining the gap in this area, and providing the hypothesis and aim of the study.
- Avoid using abbreviations in the title of your MS (e.g. PDO).
- Within the MS, and disregarding the abstract section, any abbreviations should be defined the first time you mention them. Subsequently they are allowed to be mentioned as their defined abbreviation all through the MS (e.g. PDO in the 2nd line of your MS).
- Page2-Line3: “The major criticism in modern …”. Is it “the major” or “one of the major”?!
- Page2-In the paragraph starting with “In this context, last September 2024 …”, what is DOP? All abbreviations should be defined the first time you mention them in the MS.
- It is scientific to mention the hypothesis and aim of study in the last paragraph of “introduction”.
Materials and Methods:
- Under “Animals and Feeding Thesis”, instead of bringing the rage for pigs’ weights, try to bring the Mean ± SD values, which is more scientific.
- Under “Table 1a”, please revise the “C diet, T1 diet, and T2 diet” to only “C, T1, and T2” since as you already defined them in the text, the word “diet” is redundant.
- Under “Table 1a”, as a recommendation the ranges for BWs, could be assigned by e.g. P1 (or Starter), P2 (or Grower), and P3 (or Finisher), then defining them as table foot note as P1; Period 1, ….”. this will make the table clear for the readers.
- Under “Table 1a”, the table needs a serious revision, as some of the terms are brought in Italian not English, for example Soia 44, and etc. Please revise carefully. Also, check that only the first letter should be in capitals.
- Under “Table 1a”, as a recommendation, add the units to the ingredients, (e.g. Maize, %; Barley, %, and …) in one column.
- Under “Table 1b”, Bring the ED to the top as a standard method of reporting Energy as the first composition of feed.
- Under “Table 1a and 1b”, please whether bring the full term, or in case of using abbreviations like “ED” define them in the table footnote.
- It is more suitable to mention the study period in “Animals and Feeding Thesis” section rather than “Data collection and calculation”.
- Under “Data collection and calculation”, please include the average final weight for slaughter as Mean ± SD, to be scientific.
- The data presented in table 2, is better to be reported as average daily dry matter intake (DMI) per animal, in the “Results” section and under tables reporting the performance
- Table 3 should be reported in the “Results” section. Also, as already mentioned, it is better to define live weight classes instead of bringing the numerical ranges every time.
- In the method you described as “The cold weight is obtained by deducting 2% of the hot weight of the carcass measured within the first 45 minutes of the animal's death …” a standard method of reporting cold carcass weight?
- Under “Statistical analysis”, Lines 9 to 11: You would better move this information to footnotes of corresponding tables in “Results” section.
Results:
- Under “Table 4”, remove unnecessary columns and information. For example, you would better remove the dates in the first column and use instead “Initial live weight, kg” and “Final live weight, kg”. In addition, the information mentioned in the second, third, and fourth columns are redundant here.
- Under “Table 4”, provide either of standard deviation (SD) or standard error of the means (SEM), as well as P-Values for each row.
- Under “Table 4”, provide footnotes explaining any abbreviations and any necessary information.
- Under “3.2. Nitrogen exctection and efficiency”, correct the word “Excretion”.
- Under “3.2. Nitrogen exctection and efficiency”, as you already assigned codes to any of experimental diets, you have to bring only codes in this section, not both names and codes.
- Under “Table 5”, follow a constant method of reporting units in all tables, so correct the first column to be similar to previous ones. In addition, the second column again is redundant here. Also, it is better to move the P-Values to the last columns of the table, and limit the decimals to three digits. Finally, remove the superscript “1” from next to “A” and make all the A-B letters next to values as superscript.
- Do all the mentioned modifications for table 6 as well.
- For table 7, you must include either of total or remaining fatty acids in the table, as well as units of values in “mg/100g of meat” as this is the latest accepted unit of fatty acid profile reports in meat. Same modifications as previous tables should be carried out for this table as well.
Discussion:
- You would be more focused on the mechanisms relating to the findings of your study in this section. Therefore, polish this section to be more focused.
The submitted manuscript strongly requires to be edited and rewritten by an English-language native expert. There are many errors and Italian words, especially in the tables.
Author Response
Thank you for the valuable and useful suggestions, most of which have been applied to the new version of the text, the authors' response is attached.

Reviewer 3 Report
Comments and Suggestions for Authors
It is necessary to edit the references in the paper, there are missing references, but they are in the reference list. These are references number 13, 14, 15, 16, 17, 18, 24, 38,39 and 43.
This is significant research and can be accepted after correcting the references.
13. Kerr, B.; Southern, L.; Bidner, T.; Friesen, K.; Easter, R. Influence of dietary protein level, amino acid supplementation, and dietary energy levels on growing-finishing pig performance and carcass composition. J Anim Sci. 2003, 81, 3075-3087.
14. Yi, X.; Zhang, S.; Yang, Q.; Yin, H.; Qiao, S. Influence of dietary net energy content on performance of growing pigs fed low crude protein diets supplemented with crystalline amino acids. J Swine Health Prod. 2010, 18, 294–300.
15. Tous, N.; Lizardo, R.; Vilà, B.; Gispert, M.; Font-i-Furnols, M.; Esteve-Garcia, E. Effect of reducing dietary protein and lysine on growth performance, carcass characteristics, intramuscular fat, and fatty acid profile of finishing barrows. J Anim Sci. 2014, 92, 129–40.
16. Cappelaere, L.; Van Milgen, J.; Syriopoulos, k.; Simongiovanni, A.; Lambert, W.; Létourneau-Montminy, M. Effect of reducing dietary crude protein on growth performance of fattening pigs: a meta-analysis. J. Anim. Sci. 2022, 100, 81–81
17. Lindberg, J. Review: Nutrient and energy supply in monogastric food producing animals with reduced environmental and climatic footprint and improved gut health. Animal 2023, 17, 100832.
24. Millet, S.; De Cuyper C.; Aluwé M.; Ampe, B.; De Campeneere, S. Effect of decreasing crude protein levels on the performance and carcass quality of growing-finishing pigs. Ghent (Belgium): Flanders Research Institute for Agriculture, Fisheries and Food (ILVO) 2019.
...
Author Response

(The authors gave the same response as above.)

Round 2
Reviewer 2 Report
Comments and Suggestions for Authors
After careful revaluation of the manuscript (animals-3606317), titled “Nutritional solution for the Italian heavy pig production to improve nitrogen efficiency while maintaining productive performance and meat quality.” The authors have nicely improved the MS and responded to my comments. However, I left my final comments on their MS and will leave it to the editor for final decision. Good luck.
1- For my previous comment “The “introduction” section is too long and scattered …” I still believe the “introduction” could be improved with more focused trend of writing. However, I leave it to the editor for final decision.
2- Re. previous comment “Page2-In the paragraph starting with “In this context, last September 2024 …”, what is DOP? …” The authors have made the modification; however, I suppose after mentioning the whole phrase, the abbreviation should be inserted in “()”. This is however, most into final pagination and edit stage.
Author Response
Comment 1- For my previous comment “The “introduction” section is too long and scattered …” I still believe the “introduction” could be improved with more focused trend of writing. However, I leave it to the editor for final decision.
Response 1 - The introduction is indeed a little long, but it touches on the topics of interest to us and we believe that shortening it could damage its structure. We would prefer to keep it as is.
Comment 2- Re. previous comment “Page2-In the paragraph starting with “In this context, last September 2024 …”, what is DOP? …” The authors have made the modification; however, I suppose after mentioning the whole phrase, the abbreviation should be inserted in “()”. This is however, most into final pagination and edit stage.
Response 2 - Edit